# Risk Perception towards COVID-19: A Systematic Review and Qualitative Synthesis

**DOI:** 10.3390/ijerph19084649

**Published:** 2022-04-12

**Authors:** Sabrina Cipolletta, Gabriela Rios Andreghetti, Giovanna Mioni

**Affiliations:** Department of General Psychology, University of Padua, 35122 Padova, Italy; gabriela.riosandreghetti@studenti.unipd.it (G.R.A.); giovanna.mioni@unipd.it (G.M.)

**Keywords:** coronavirus, COVID-19, health, pandemic, risk perception, systematic review

## Abstract

Several studies have attempted to identify how people’s risk perceptions differ in regard to containing COVID-19 infections. The aim of the present review was to illustrate how risk awareness towards COVID-19 predicts people’s preventive behaviors and to understand which features are associated with it. For the review, 77 articles found in six different databases (*ProQuest*, *PsycInfo*, *PubMed*, *Science Direct*, *SCOPUS*, and *Web of Science*) were considered, the Preferred Reporting Items for Systematic Reviews and Meta-Analyses (PRISMA) guidelines was followed, and data synthesis was conducted using a mixed-methods approach. The results indicate that a high-risk perception towards COVID-19 predicts, in general, compliance with preventive behaviors and social distancing measures. Additionally, risk awareness was found to be associated with four other key themes: demographic factors, individual factors, geographical factors, and timing. Therefore, gaining a greater understanding of individual and cultural differences as well as how people behave could be the basis of an effective strategy for raising public risk awareness and for countering COVID-19.

## 1. Introduction

In December 2019, the Wuhan Municipal Health Commission in China confirmed a series of pneumonia cases in Wuhan, Hubei province’s largest city. The cause of the pneumonia cluster events was unclear, and no deaths were reported. A few days later, Chinese authorities informed the World Health Organization (WHO) of the presence of a new infectious respiratory disease, provoked by a novel coronavirus (SARS-CoV-2). In total, 7818 coronavirus cases were confirmed by the end of January, with 82 of them occurring in 18 different countries. COVID-19 was declared a global pandemic by WHO approximately one month later, following a massive rise and spread of infections [1].

The novel coronavirus gives rise to a series of physical symptoms, which range from mild to extreme. Fever, fatigue, and cough are considered common symptoms, and serious manifestations may include dyspnea, pneumonia, and acute cardiac injury [2]. Aside from physical symptoms, contracting the new coronavirus or being isolated to protect oneself from it can provoke a string of mental health implications, such as anxiety, depression, and dementia [3].

Hence, it is clear that researchers have been attempting to decipher which factors can improve the public’s general wellbeing during the pandemic [4] and the characteristics that are involved in the development of psychiatric complications [5]. Additionally, they are seeking to understand how people’s perceptions regarding COVID-19 can impact their behaviors towards it and why those perceptions vary according to both individual and cultural differences [6].

Risk perception enters into the picture not only as a factor that affects people’s mental health but also as a tool that helps them avoid contracting the new coronavirus [7]. While it is not surprising that a pandemic could cause both physical and mental health problems, knowing how individuals view COVID-19 and defend themselves from it can provide us with useful knowledge about how to control the outbreak. Research has shown that people’s risk perceptions of COVID-19 have a significant impact on how they manage their mental wellbeing during the pandemic [8] as well as whether or not they are protecting themselves and practicing preventive behaviors [9].

The aim of this review was to analyze how risk perception towards COVID-19 impacts preventive behaviors. Furthermore, the current study seeks to determine which personal characteristics are linked to risk awareness and if there are any mediators between risk tolerance and protective behavior engagement.

## 2. Materials and Methods

A systematic literature review was developed following the Preferred Reporting Items for Systematic Reviews and Meta-Analyses (PRISMA) guidelines [10].

### 2.1. Search Strategy

All the articles presented in this review were collected from six different databases: *ProQuest*, *PsycInfo*, *PubMed*, *Science Direct*, *SCOPUS*, and *Web of Science*. Key words such as “risk perception”, “COVID-19”, “risk awareness”, and “coronavirus” were used in combination for the selection. Most studies were found using the key terms “risk perception” and “COVID-19” together.

### 2.2. Study Selection and Inclusion Criteria

Titles and abstracts of each publication were screened for relevance. Full-text articles were accessed for eligibility after initial screening. The inclusion criteria were: (1) studies conducted during the first pandemic year, specifically between March 2020 and February 2021, (2) that assessed COVID-19 risk perception and its associated factors and (3) that were written in English.

As presented in the PRISMA diagram (Figure 1), 1251 articles were identified in the six databases. However, only 398 were found to be specifically related to risk perception and COVID-19. Among them, 292 articles were excluded because they did not report any studies, or their abstracts contained either “COVID-19” or “risk perception” and its variants but not both terms. Based on a review of full texts, another 29 articles were excluded for the following reasons: nine did not report studies, seventeen did not assess risk perception, and three had no full text available. Hence, in total, 77 articles were considered for the present study.

### 2.3. Data Extraction

The entire systematic review process was conducted by one independent reviewer who collected the articles and analyzed each one of them carefully. Papers were incorporated into the study according to the inclusion criteria. A second reviewer re-analyzed the report, advising the first one on which variables should be added and omitted from the review, as well as checking the accuracy and quality of the data. Finally, the third author offered a third point of view to increase the accuracy and quality of the analysis.

Data synthesis was conducted by summarizing the most relevant information from the articles, as reported in Table A1 (reported in the Appendix A). The table was constructed by gathering data from each paper: authors, country, sample, data collection methods, risk perception measures, and key findings. It was designed not only as a way of clearly displaying data for this analysis but also as a manner of providing efficiently synthesized information.

Following analysis of the table, emergent themes were identified, and due to the diversity of articles presented in this review, a mixed-methods approach was selected as a strategy for adequately reuniting qualitative and quantitative data in a systematic review [11]. The quantitative findings were converted into qualitative themes, which were then pooled together to identify the main themes and sub themes.

Quality appraisal was conducted following the Joanna Briggs Institute (JBI) critical appraisal checklist [12], which is composed of eight items assessing each study on the basis of its sample inclusion criteria, sample description, the validity and reliability of the measures; the use of objective and standard criteria, the identification of confounding factors and of the strategies to deal with them, the validity and reliability of the outcome measures; and the appropriateness of statistical analysis.

## 3. Results

Findings from 77 academic studies were selected for this review and are summarized in Table A1. Almost all studies (76 out of 77) followed a cross-sectional modality except for one [13], which opted for a combined cross-sectional and longitudinal survey. Most data collection methods contained questionnaires, scales, and/or interviews, and only one study [14] deployed qualitative data collection via social media posts and emails. Additionally, the majority of studies used cross-sectional questionnaires with closed-ended questions; four [14,15,16,17] exclusively looked at qualitative data (e.g., phone interviews with open-ended questions), and two chose a combination of open-ended and closed-ended questions [18,19].

Quality appraisal showed that the first and second items of the JBI checklist regarding sample inclusion criteria and description were satisfied in all the studies but one [14], where the two items are not applicable because the study used a thematic analysis of emails and social media messages. In addition, every study met the requirements for measurement validity and reliability (third item). As for the use of objective and standard criteria (fourth item), it is possible to declare that standard criteria were used by all studies, but the measurements are not objective, as all of them utilized the self-report approach, which can lead to phenomena such as the social desirability effect and dishonesty. Regarding the fifth and sixth items, the identification of confounding factors (e.g., cultural characteristics, religion, and the misrepresentation of other factors) was frequently disclosed in the studies’ limitations and described as a means to improve future research.

With respect to the validity and reliability of the results (seventh item), it is important to note that, due to the COVID-19 pandemic situation, most of the studies (70 out of 77) used online data collection methods; thereby, the results exclude or underrepresent uneducated, unprivileged, as well as older people. As a result, despite the efforts to represent the general population or specific groups (e.g., health care workers), the findings cannot be blindly generalized to the whole population. Furthermore, because 76 studies were designed using the cross-sectional approach, causal inferences are impossible to draw, and it could leave space for recall bias. Finally, every study matched the last item regarding the appropriateness of statistical analysis, with the exception of qualitative studies where this criterion was not applicable.

As a result of this research synthesis, five major themes were found to be correlated with risk perception: engagement and compliance with preventive behaviors, demographic factors, personal factors, geographical factors, and timing.

### 3.1. Engagement in Preventive Behaviors and Compliance with Government Recommendations

According to the results, the analysis of risk perception is critical for both recognizing how it impacts people’s mental health and for exploring its involvement in government-suggested preventive behaviors. In several studies, high risk awareness predicted high compliance with government rules and social distancing measures [10,20,21,22,23,24,25,26,27] as well as having a great impact on people’s feelings and daily habits [27].

Information seeking, respect for government rules, and frequency with which people participate in behaviors and activities that can enhance their wellbeing and hygiene were all seen to profoundly change during the pandemic. High risk perception stimulated feelings of fear and anxiety among the general public [27] as well as expanded engagement in COVID-19 knowledge seeking [28].

Nevertheless, it is not only risk perception that has a direct impact on COVID-19-protective conduct. Government guidelines per se were found to be one of the most influential factors for the adoption of disease-preventive behavior [29], as they can also alter risk tolerance. These data are consistent with the findings of a study [30] in which 71.3% of the sample reported that government guidance influenced their behavioral change during the pandemic.

It is important to acknowledge, however, that compliance can be expressed differently in public or in private. In a study conducted in Germany, only a quarter of the sample expressed full adherence to COVID-19 recommendations, while more than a half (51%) intended to follow some public actions but were less willing to enact personal hygiene behaviors [31].

A minority of studies contained in this review found little or no impact of risk perception on compliance with COVID-19-preventive behaviors. In one study [32], for example, even though the sample generally presented a low risk perception, it still supported restrictive policies with preventive intent. Furthermore, another one [33] outlined a negative correlation between risk tolerance and COVID-19-protective engagement.

### 3.2. Demographic Factors

Thirty-one studies pointed out that demographic factors produce a significant impact on COVID-19 risk perception and preventive behaviors. Age, gender, income, employment, and education were examined individually as subcategories.

#### 3.2.1. Age

Four studies indicated that being older was associated with both higher risk perception [20,34,35,36] and two [37,38] with engagement in preventive behaviors. Younger age, nonetheless, sometimes predicted high risk perception [31,39] and compliance with government policies to prevent spread of the new coronavirus [31]. Italian young adults, for example, rated the pandemic as severe and showed a high risk awareness [8].

The results of three studies indicated that adolescents and young adults present a higher risk perception for others and for their relatives than for themselves [40,41,42]. As a result, appeals to altruism can be seen as a potential strategy for improving coronavirus-preventive measures among teenagers and young adults [42].

Additionally, two studies identified contrasting age ranges within which their participants indicated the highest risk perception: being over 45 years old [43] and between 20 and 30 [44]. However, it is important to highlight that these studies were conducted in two distinct countries (Ethiopia and Indonesia) with very different cultures.

#### 3.2.2. Gender

Female gender presented the highest association with risk perception regarding COVID-19 [24,31,34,35,41,45] in the studies in which this factor was analyzed. Moreover, being a woman was also a predictor of engagement in COVID-19-protective and -preventive practices [23,38,46]. This pattern was found in several countries [45], and in all of them (Australia, Germany, Italy, Japan, Mexico, UK, USA, South Korea, Spain, and Sweden), being male was linked to lower risk awareness.

Nevertheless, three studies [36,47,48] found male sex was associated with a higher risk perception score, and one [49] pointed out that being a man was linked to a greater likelihood of getting vaccinated in the future.

#### 3.2.3. Income

Five studies directly explored the impact of income on COVID-19 risk perception [15,16,30,44,50], and all of them came to the conclusion that higher overall earnings were positively correlated with it. On the contrary, low risk perception, a lack of protective and preventive practices in the face of the pandemic, and non-adherence to social distancing measures were all observed in poor communities with restricted access to high-quality health services.

In a study conducted in India [15], for example, most participants thought they had little or no chance of personally contracting the virus, and in two studies [16,50] conducted in the United States with Latino communities, a direct link between poverty and low COVID-19 risk perception was found. The vast majority of Latino people did not participate in preventive behaviors [50], and the results indicated that these groups may be at higher risk of contracting coronavirus due to factors such as limited access to high-quality health care and low incomes [16]. Moreover, according to the findings, Latino people show a low COVID-19 infection risk awareness because they are more concerned with other possible consequences of the pandemic, such as pay cuts or job losses.

#### 3.2.4. Employment

The healthcare profession was the focus of many studies that aimed to understand how people on the front lines and in direct contact with coronavirus feel and behave. Furthermore, it is relevant to comprehend how they perceive it and if there are differences between them and the general population. Three studies [44,51,52] pointed out that healthcare workers presented a higher risk perception for COVID-19 and adopted more protective behaviors than the general public, and only one study [36] found that healthcare workers were barely concerned about coronavirus infection or only occasionally alerted to the virus.

Among healthcare workers, the nursing profession was found to be even more associated with a higher risk perception of coronavirus infection [53]. Indeed, according to a study carried out in Spain [54], 37.5% of the nurses in the sample were afraid of contracting the virus and of its possible consequences, and 62.8% feared infecting people they lived with. This concern about being infected while working was found not only among healthcare professionals but also in other workers who had a job outside the home [34]. Additionally, the pharmaceutical profession was related to higher COVID-19 threat perception scores [48].

#### 3.2.5. Education

High educational level, university enrolment, and possession of a college degree were all predictors of increased COVID-19 risk perception [13,39,41,43] as well as higher compliance with behavioral guidelines to contain infection spread [13,31].

These results, however, do not entirely correspond with others found in this review. In three studies [33,38,47], being more educated was linked to greater involvement in protective and preventive behaviors but not to risk perception. This may be because higher education might help people to engage in safety behaviors but at the same time protect them “from a (possible) irrational fear of being infected or dying” [38].

### 3.3. Personal Factors

In addition to demographic factors, personal characteristics were shown to have a significant weight with regard to COVID-19 risk perception. The sub themes identified in this review were: health status, mass media influence, COVID-19 knowledge, wellbeing, political orientation, trust, personality and conspiracy mentality, optimistic bias and positivity, direct and indirect experience, and propensity to vaccinate.

#### 3.3.1. Health Status

Health status is a delicate topic to investigate in the context of a pandemic. Six of the studies presented in this review [17,23,34,42,55,56] fully captured this sub theme and its relationship with risk perception.

The main outcomes reveal that both individuals with chronic conditions [17,42,56] and people that perceive their own health as poor [23,34] present higher risk perception scores. However, in one American study [55], COVID-19 fatality risk was underestimated among individuals with pre-existing medical conditions.

#### 3.3.2. Mass Media Exposure

Six studies [24,27,35,48,57,58,59] have shown the influence of mass media exposure on coronavirus risk perception. According to one study [58], 50% of research participants reported that their COVID-19 risk awareness changed after reading or hearing about it in the media, and this information is in line with the findings of two other studies [35,48], which pointed out that being constantly exposed to COVID-19-related messages not only influenced risk perception but also increased perceived uncertainty and acceptance of mitigation measures. Nevertheless, as stated by one of the studies [59], not only the amount of time spent consuming COVID-19 media content but also how one receives this information is associated with higher risk perception.

In only one of the examined articles [60] was time spent consuming information through news, social media, or health websites not connected to risk perception.

#### 3.3.3. COVID-19 Knowledge

COVID-19 knowledge was the most explored subtheme in the papers contained in this review. Due to the extensive coverage of this topic, distinct results were found. Ten studies [18,20,26,37,42,43,61,62,63,64] observed a correlation between coronavirus understanding and risk perception, but other matters, such as how COVID-19 misconceptions cause irrational risk beliefs in people [65,66] and how knowledge regarding it affects preventive behaviors [19,29,33,64], were also investigated.

Two studies reported risk perception to be influenced by personal knowledge about COVID-19 [61,64], six studies found a correlation between these factors [18,20,26,42,43,63], and one paper claimed that high risk perception and uncertainty nurture coronavirus-information seeking [62]. In the first two articles mentioned above [61,64], high levels of knowledge seeking provoked higher levels of risk perception, but the parameters were not so consistent among studies.

In a study conducted in Nigeria [18], only 26% of participants knew they could contract COVID-19, and in fact, 12% affirmed the pandemic was an “exaggerated event”, indicating that people with poor COVID-19 awareness underestimate risk. These results are consistent with the ones proposed by other studies [20,26,42,63]. Only two studies detected no correlation between knowledge about the virus and risk perception [36,38], and only one found that poor knowledge was associated with higher risk perception [43].

Misconceptions were a topic mentioned in two studies. The first one [65] reported several irrational beliefs among participants, including the thought that East Asian individuals show higher levels of COVID-19 infection. In the other study [66], most participants thought there was a higher probability of COVID-19 fatality among older people and people with chronic health conditions but not among lower income and Black participants. Hence, economic and racial disparities were not clearly seen by these participants.

Finally, knowledge of COVID-19 was a predictor for embracing safety measures in almost all studies on the topic [29,33,37,64], and only one study developed in Greece [19] showed that while many respondents had a thorough understanding of COVID-19 transmission modes and prevention strategies, good practices were not reported at the same level.

#### 3.3.4. Wellbeing

COVID-19 risk perception was negatively correlated with wellbeing in nine studies [7,8,27,67,68,69,70,71,72], and most claimed that scoring high for COVID-19 risk awareness predicted fear, anxiety, and stress [7,27,69,72] as well as psychological disturbance [8,67]. In a study conducted with Italian parents [72], for instance, risk awareness was linked with parents’ stress levels and children’s psychological problems. Moreover, affective risk perception was positively associated with depression [67].

Distress, apprehension, and anxiety were associated with higher risk perception [70,71]. Indeed, according to a Spanish study [70], participants who experienced tension increased their risk awareness levels. Furthermore, one analysis found that although high risk perception is negatively correlated with wellbeing, it enhances coping strategies that could help control the pandemic [68]. Finally, a lower degree of perceived social support was linked to higher levels of active coping with COVID-19, and this association was mediated by high risk awareness [73].

#### 3.3.5. Political Orientation

Two American studies [74,75] presented an interesting link between political orientation and COVID-19 risk perception. According to one of them [74], Democrats perceived more risk associated with COVID-19 than Republicans. A higher perceived risk of infection, fatality, possible disease outcomes, and financial ruin were all observed more frequently in the Democratic party as well as a deeper engagement in preventive measures. As stated by the authors, perceptions of risk may be socially constructed and differ based on political inclinations. Hence, understanding political preferences in a pandemic context could be used as an effective strategy for containing the outbreak.

Additionally, the other study [75] explored the notion of “echo chambers” in order to investigate how COVID-19 risk perceptions may vary across preferences for conservative or liberal bias. Indeed, risk awareness showed a variation between these two biases, but the “echo chambers” phenomenon was not supported by the research. Moreover, people presenting a low information-seeking behavior were more prone to believe in politicized COVID-19 news. Consequently, political orientation and political spread of news may influence risk perception towards COVID-19.

#### 3.3.6. Trust

Four different sorts of trust were evaluated by studies: government trust, general trust, social trust, and media trust. General trust and social trust showed unanimous effects in two studies [76,77]. In both, people with general trust perceived less risks associated with COVID-19, whereas social trust indicated higher risk awareness scores.

In another study [45], risk perception was found to be negatively affected by government confidence in two of the ten countries presented in the analysis (South Korea and Spain), and no correlation was found in the others. However, other research [58] pointed out that when people trusted the authorities, they were more likely to engage in COVID-19-protective behaviors. Trust in local government and media helped to reduce infection rates of diseases in another study [77], and risk perception either entirely or partly mediated the effects of the kinds of trust presented in it.

#### 3.3.7. Personality and Conspiracy Mentality

Only one of the examined articles [78] explored the association between personality and risk perception and did not find significant results. Nonetheless, scoring low on agreeableness (as defined by the Big Five model) and high on aspects of the Dark Triad was associated with less compliance with COVID-19 restrictions. Another study [79] analyzed the weight that conspiracy mentality may have on risk perception and found that this phenomenon was correlated with a higher risk awareness of death but not with other types of risk. Moreover, people that presented a conspiracy mentality were more likely to adopt mitigation behaviors when they perceived a risk to themselves.

#### 3.3.8. Optimistic Bias and Positivity

Optimistic bias was found to have a direct effect on COVID-19 risk perception in four studies [13,28,50,80], and all studies’ results indicate that these two variables are negatively correlated.

As maintained by one researcher [28], optimism affects COVID-19 risk awareness directly and engagement in safety behaviors indirectly because risk perception enhances risk response. This finding is consistent with another study [50] in which participants who presented optimistic bias had a lower risk perception and were less likely to take preventive measures.

In addition, another research [8] claimed in their analysis that the influence of COVID-19 perceived risk on death distress and happiness was mediated by positivity in the sense that an optimistic person is more able to develop coping strategies aimed at reducing psychological stress and enhancing wellbeing. As a result, being positive was associated with lower COVID-19 risk perception but increased happiness and serenity.

#### 3.3.9. Direct and Indirect Experience of COVID-19

Direct experience of COVID-19 was a great indicator of elevated risk awareness in seven studies [34,41,44,48,53,71,81]. In one [53], healthcare workers felt themselves more at risk of contracting the virus than their family members, and in another [81], physicians and nurses presented higher risk perception scores than other healthcare professionals, especially because they usually get closer to patients.

Personal location also influenced the public’s perception of direct and indirect contact with COVID-19. Two studies [44,48] found that people living in urban cities and residents of Wuhan city or Hubei province had higher COVID-19 risk perception [41,71].

#### 3.3.10. Propensity to Vaccinate

As all articles contained in this review were published in 2020 or at the beginning of 2021, no vaccines for combating COVID-19 were yet available, and few studies were concerned with the issue of how many people intended to get vaccinated in the future or its relationship with COVID-19 threat perception. Nevertheless, all studies addressing this issue indicated that higher risk awareness was associated with greater inclination to be vaccinated [9,24,49,59].

A Chinese study [49] showed that 91.3% of respondents claimed they would accept COVID-19 vaccination, and having a higher COVID-19 perceived risk was associated with this attitude.

### 3.4. Geographical Factors

The studies included in this review were conducted in 30 different countries (Australia, Bangladesh, Bolivia, Canada, China, Cyprus, Ethiopia, Finland, France, Germany, Ghana, Greece, India, Indonesia, Iran, Italy, Japan, Jordan, Mexico, Nigeria, Poland, Portugal, Spain, South Korea, Sweden, Switzerland, Taiwan, Turkey, United Kingdom, and United States), and risk perception scores varied among them. Given that the studies carried out in Canada, Cyprus, France, Greece, Jordan, Poland, Portugal, and Taiwan did not represent the general population or did not mention COVID-19 risk perception scores, they were not discussed within this theme.

A low perceived risk of infection was reported by participants in Australia, with only 5% claiming they would have a very high risk of acquiring the virus [58] and with 46.7% reporting they would expect only moderate symptoms in case of infection [9]. Low risk awareness values were also encountered in studies carried out in India [15], Ethiopia [43], Nigeria [18], Indonesia [44], and China [49,69].

On the contrary, high levels of COVID-19 risk awareness were found in Bangladesh [82], Finland [14], and Ghana [83]. Studies conducted in Iran [84], Italy [38], Turkey [8], Switzerland [76], and Bolivia [24] showed moderate levels of risk tolerance among their populations.

Contrasting results were found in the United States. For example, a representative national sample of the American population [39] revealed an overestimated perceived mortality risk of COVID-19 (14%), which strongly contradicts the findings of another study [55], which reported that the perceived personal fatality risk of the respondents was lower than 1%. Additionally, other two analyses reported median risk perception scores for infection [21,32] and fatality [21] among their participants. Finally, studies conducted with some minority groups in the USA presented very low values for fear of COVID-19 infection [16,50] but a higher financial risk perception [16].

Only three articles [45,85,86] reported comparative findings among countries. In one of them [45], 10 samples from different countries were explored (Australia, Germany, Spain, Italy, Sweden, Mexico, Japan, South Korea, United Kingdom, and United States), and the most notable outcomes revealed that levels of risk perception regarding the virus were higher in the United Kingdom than in the other countries. This information is consistent with the results of another study [85], which found that the English and Spanish populations presented higher anxiety about the possibility of being infected and the severity of infection than Japanese people. Moreover, in a study conducted in Germany [86], ethnic minorities curiously showed greater health and financial risk perceptions than the general population, expressing concern regarding the consequences of the pandemic.

### 3.5. Timing

Four studies [13,21,47,55] mentioned the role of timing in influencing COVID-19 threat perception, and three of them [13,47,55] sought to compare how COVID-19 risk awareness changed across time. One [55], for example, compared the results of two cross-sectional surveys conducted in March and April 2020, respectively, and found a higher risk perception in the second survey. The only longitudinal study [13] found on the topic indicated that higher risk perception scores led to more engagement in protective behaviors.

Only one study [47], after comparing the results of two cross-sectional online surveys, established that there was a decrease in perceived risk perception scores between early and late lockdown. Interestingly, the previous mentioned studies [13,55] were all conducted in the United States, whereas this one [47] was in Bangladesh.

## 4. Discussion

This review has identified five themes related to risk perception towards COVID-19: engagement in preventive behaviors and compliance with government recommendations, demographic factors, personal factors, geographical factors, and timing. These themes were analyzed in relation to the four main kinds of COVID-19 risk perceptions the studies referred to: infection, fatality, affective, and financial. Infection threat perception was the one that was most extensively highlighted by the studies and the one that presented most correlations with the sub themes. Fatality risk awareness was raised in a few studies and was related to the following sub themes: health status, political orientation, personality, and conspiracy mentality. Affective risk tolerance was addressed especially in the studies conducted with young adults and adolescents, who were more concerned about COVID-19 infection and its consequences for their families than for themselves. Finally, financial risk awareness was present mainly in the studies conducted among low-income communities, in which fear of losing one’s job or enduring a pay cut in consequence of the pandemic was higher than concern about being infected with coronavirus.

As regards each of the themes identified in the review, the findings of the studies considered revealed that increased COVID-19 risk awareness is associated with increased engagement with preventive measures as well as more acceptance of guidelines that the government proposes. This could be because high risk perception stimulates feelings of fear and anxiety among the general public [27], which could consequently make people engage in protective behaviors and comply more with government recommendations, but this result also depends on the cultural attitude towards collective institutions. For instance, in China, people tend to be more obedient to government guidance and believe more in its recommendations [29].

Further studies [87] have pointed out that strictest containment policies are related to lower adherence, and this can be explained in terms of a decrease of self-regulation when government rules are not underpinned by sufficient information dissemination and public engagement. The health action process approach [88] suggests that risk perception, positive outcome expectancies, and self-efficacy predict intention, but a crucial mediating role between intention and action is played by action plans. This model has been shown to predict physical distancing and handwashing behavior during the pandemic [87,89] and has also been applied to the roles of habit, social norms (e.g., to protect the vulnerable people), and anticipated regret (e.g., to infect them). Another important component of the intention to follow anti-contagion measures [90] is health threat appraisal, which refers to the belief that a problem has serious negative consequences and the belief in being susceptible to those consequences.

Among the studies included in the review, 18 out of 77 [22,25,26,28,29,37,40,45,46,57,60,61,62,68,76,77,78,84] referred to a theoretical framework to explain their results on risk perception. Of them, four [22,28,37,57] used the health-belief model [91] and three [26,28,29] the theory of planned behavior [92]. Other theories used as theoretical frameworks in studies [22,29,62,68,76,77,78,84] the following: the Beck’s risk society theory [22] [93], the theory of reasoned action [29,94], the uncertainty management theory [62,95], the uncertainty in illness theory [62,96], the protection motivation theory [68,84,97], the conservation of resources theory [68,98], the theory of generalized trust [76,77,99], the Big Five theory of personality [78,100], the Dark Triad theory [78,101], and the theory of bounded rationality [84,102]. Moreover, models were also used as theoretical frameworks in ten studies [22,24,25,40,45,46,60,61,62,84]. These models are: the protective action decision model [22,103], the tripartite model of risk perception [25,104], the multidimensional model of individualism–collectivism [40] [105], the van der Linden’s model of risk perception [45,106], the extended parallel process model [46,60,107], and the model of risk information seeking [62,108].

As for demographic factors, five subcategories were extensively analyzed: age, gender, income, employment, and education. Findings indicate that being older, female, possessing a higher income, being a healthcare professional, and studying in a university or owning a college degree were all predictors of higher risk perception levels. According to a study [20], higher age is positively correlated with high COVID-19 risk perceptions (infection and fatality) because older people are more susceptible to the negative consequences of the virus. Moreover, another study [41] raised the fact that females are usually more sensitive than males, and this could contribute to perceiving COVID-19 as a higher risk. Finally, being young was associated with a higher risk awareness for family members and others than for oneself, suggesting that even though some groups believe they are not at high risk for COVID-19 infection, they are nevertheless anxious about their loved ones being infected. Finally, lower income was related to a higher risk perception in terms of finances and employment rather than infection and fatality, probably due to the life situation of people with lower incomes, which makes them fear more the financial consequences of the pandemic than other effects.

Personal factors included 10 different sub themes: health status, mass media exposure, COVID-19 knowledge, wellbeing, political orientation, trust, personality and conspiracy mentality, optimistic bias and positivity, direct and indirect experience, and propensity to vaccinate. Results indicate that perceiving one’s own health as poor and having a chronic condition were, in general, linked to higher risk awareness scores, as was greater exposure to COVID-19-related news. In particular, people with serious medical issues had higher fatality risk perception. and this might be due to their being more prone to COVID-19 negative repercussions. Furthermore, high proportions of coronavirus knowledge seeking provoked higher levels of risk perception and engagement in protective behaviors. As pointed out in a review exploring the role of social media in the general population’s risk awareness [109], media exposure has a great impact in molding people’s risk perceptions. During the COVID-19 pandemic, people were bombarded with a large amount of information, which some authors [110] have defined as an emerging infodemic. Previous studies [111,112] associated infodemic with a decrease in psychological well-being, and one study in this review [24] explained the association between media exposure and risk perception referring to the mediating role of two self-relevant emotions (fear and anger) that influence risk perception and the subsequent adoption of protective measures [113].

Wellbeing was negatively associated with COVID-19 risk awareness, as higher risk perception predicted fear, anxiety, and stress. In addition, being positive and having unrealistic optimism [95] were negatively correlated with threat perception. People believing that their personal outcomes would be more favorable than others’ during the pandemic not only did not fear coronavirus but also were not properly engaged in the recommended safety behaviors. Nevertheless, positivity was associated with coping strategies aimed at enhancing one’s mental wellbeing and reducing stress. Thereby, it may be inferred that a positive attitude may help to feel better, especially in the short term, but may lead to undesired consequences, especially in the long term, if it is associated with unrealistic beliefs. Knowledge may play an important mediating role in this process [114].

Different types of trust (general, social, government, and media trust) were associated with COVID-19 risk perception. Findings show that general trust presents a negative correlation with risk awareness, and social trust a positive one. One explanation for the negative correlation between general trust and COVID-19 risk perception might be that people who are more likely to trust others unconditionally are also more likely to be optimistic about the risk of hazards [76,77]. In the case of social trust, its positive relationship with risk perception can be explained by the fact that in situations where people have little knowledge about hazards, they must rely on institutions to appraise the hazards’ risks. In other words, people’s COVID-19 risk perceptions are influenced by institutional knowledge that shapes their ideas, and social trust and risk perception are thus positively associated [77].

Government trust was associated with lower risk perception in only two of the eleven countries where it was explored in a study [45]. This difference among countries might depend on the varying government measures applied, which may have a more or less reassuring effect. It might also be that other factors play a more salient role. Trust in the media was associated with higher risk awareness and infection rates in one of these countries [77], thus pointing out the importance of trustworthy media communication [14].

Other personal factors, such as having direct experience or exposure to the virus (as in the case of healthcare professionals and people living in urban cities) and propensity to vaccinate, were associated with higher risk perception. The results show that the people who are more exposed might be the ones who are more concerned about the virus and more interested in protecting themselves from it. By way of explanation, high risk perception can increase one’s tendency to protect oneself. These data are supported by recent studies in which vaccination uptake was reported more positively by healthcare professionals when compared to people in other professions [115].

Few articles explored the role of political orientation or personality and conspiracy mentality in risk perception. The studies that addressed politics as their main theme were American and found out that Democrats perceived more risks associated with COVID-19 than Republicans as well as a greater engagement in preventive behaviors [74]. These data could be explained by Republican leaders’ desire to reopen the American economy in April–May 2020 as well as the varied news sources used by members of the two parties [74].

Furthermore, people presenting lower COVID-19 information-seeking behavior were more prone to believe in politized coronavirus news as opposed to high information seekers who use a variety of platforms and are more open to information that contradicts their current ideas or inclinations [75]. As for personality traits and conspiracy mentality, these did not significantly influence risk perception because dispositional tendencies override severe situations such as a pandemic, whereas conspiracy mentality was only positively associated with fatality risk perception. Because possessing a conspiracy mentality generally means having an aversion to political guidance, this group will only see COVID-19 as a risk if they perceive their lives to be in danger [79]. As a result, this set of people, who may be hesitant to seek political advice on a regular basis, may have a change of heart when confronted with a critical situation.

The fourth theme presented in this review was geographical factors. As the studies come from 30 different countries, changes in risk perception and how people react during the pandemic could be attributed to where they are from or what their cultures are like. Differences in risk awareness were established between countries: Australia, India, Ethiopia, Indonesia, Nigeria, and China showed low risk perception scores, and on the contrary, high levels of COVID-19 risk awareness were found in Bangladesh, Finland, Ghana, and the United Kingdom. Additionally, other states such as Iran, Italy, Turkey, Switzerland, and Bolivia reported moderate values for risk tolerance. As the countries that show similar levels of risk perception come from very different parts of the world and exhibit distinct financial conditions, cultures, and religions, it is not simple to deduce the reason they are alike on this parameter. Apart from Australia, the nations with low risk perception scores are low-income or developing countries, and the fact that these populations may not be as aware of the virus’ potential consequences may cause them to be less concerned. As regards Australia, unlike the other identified countries, where there was a low risk perception, it was not severely affected by coronavirus consequences.

Countries with medium and high levels of COVID-19 risk awareness appear to have little in common, making it hard to pinpoint the cause of the similarity. Nevertheless, hypotheses about the risk perceptions of some nations can be derived. Coronavirus wreaked havoc in the United Kingdom, for example, resulting in the highest death toll in Europe [116]. Furthermore, this sovereign country also experienced a series of lockdown periods [117], which may have heightened people’s perceptions of COVID-19 as a threat. As for Finland, low government trust seemed to be the most probable cause of COVID-19 high risk awareness levels [14], increasing the population’s fear towards the virus. Strangely, even though Italy was one of the most affected countries in the first lockdown period (March 2020), results still indicated medium levels of COVID-19 risk perception among participants of Italian studies. In one study [38], for instance, threat perceptions were higher for finances and employment and lower for health. This may indicate that Italians were more concerned about the negative financial consequences of the pandemic than the health ones. Moreover, Italy was one of the countries to impose the earliest and strictest anti-contagion measures, which might have had a protective role in mitigating anxiety and reducing risk perception [118,119].

Many studies were conducted in the United States with varying results for infection and fatality risk perceptions among the general population. However, research focusing on Latinos living in the United States [16,50] showed that these individuals have lower COVID-19 threat infection perception but are more fearful of losing their jobs or having their salary slashed. These data are in line with trends in developing or low-income countries and may be interpreted again as a result of the worse economic situation of this ethnic minority compared to the general population. Other countries not listed in this theme either did not use general population samples or did not utilize COVID-19 risk awareness scores.

Timing was the last theme discussed in this review, and most articles that considered this dimension came to the agreement that COVID-19 risk perception scores increased with the passage of time. Nevertheless, it must be kept in mind that three of the four articles analyzing this dimension were carried out in the United States in a period (March–May 2020) when contagion rates were rapidly increasing there [117], and this might have resulted in a higher risk perception.

### Limitations

The present review contains a series of limitations. To begin with, all the studies gathered were published in English, therefore excluding high-quality analyses written in various other languages. Second, the number of studies on COVID-19 risk perception is growing, making it difficult to acknowledge whether the findings in this review are consistent with more recent studies. Finally, the bottom-up approach followed in this review and the variety of the variables considered in the different studies prevented us from grouping the results in a unified theoretical model, which is able to synthesize the disparate findings.

## 5. Conclusions

In conclusion, the results of this review showed that a higher COVID-19 risk perception implied more engagement in preventive behaviors and more compliance with government recommendations. Being older, female, having a higher income, being a healthcare professional, and studying in a university or owning a college degree were all predictors of a higher risk awareness. Additionally, with regard to personal factors, having a chronic condition or perceiving one’s health as poor, being constantly exposed to COVID-19-related news, seeking information linked to the virus, having direct contact with it, social trust, propensity to vaccinate, and being a Democrat all presented a positive correlation with risk perception.

Even though studies were conducted in 30 different countries, the only differences derived from their comparison are that people from low-income/developing countries and from countries with lower rates of contagion reported lower risk perception. Moreover, risk awareness was observed to increase in the middle of 2020 compared to the beginning of the same year, implying a rise in concern as the number of cases climbed throughout the pandemic outbreak.

Therefore, gaining a greater understanding of COVID-19 risk perception and how it varies according to individual and cultural differences could be used as an effective strategy for raising people’s engagement in preventive and protective behaviors as well as for countering the virus. However, it is crucial to note that the pandemic was in a different stage when this review was written than when most of the studies contained in it were conducted, and the actual distribution of the vaccines will almost certainly have had a significant impact on public perceptions of COVID-19 threat. Thus, further and more detailed research is needed to fully comprehend this phenomenon.

## Figures and Tables

**Figure 1 ijerph-19-04649-f001:**
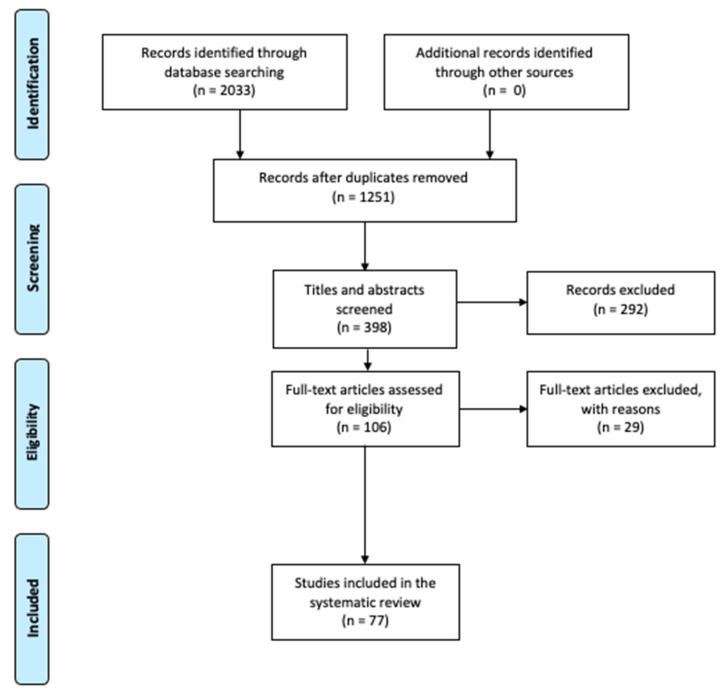
PRISMA flow diagram for study selection.

## Data Availability

Not Applicable.

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
