# Peer review of "Risk Perception towards COVID-19: A Systematic Review and Qualitative Synthesis"

_ijerph, 2022, doi:10.3390/ijerph19084649_

Round 1

Reviewer 1 Report

Conducting the systematic reviews is the useful method to  grasp the variety of findings on important and “hot” topics. As Covid-19 pandemic was subjected to extensive research all efforts made to clarify and organize findings form numerous studies warrant attention.

The presented manuscript is one example of such effort. What is more the Authors decided to use PRISMA guidelines in their analyses.

I found the manuscript both interesting (due to the effort of performing a review) and difficult to read and get a clear picture of Authors’ findings. It might be due to some aspects of the way the analysis was conducted and the findings presented. Below are the most important – in my view – weaknesses of the manuscript.

  1. Inclusion/exclusion criteria of papers should be described in more details. The phrase “poorly addressed coronavirus risk awareness” as the exclusion criterion is not clear and does not provide information on what aspects of risk assessment were considered. Thus part 2.4 should be elaborated as in the present form it does not explain precisely what criteria were used. It is not clear from the description and Diagram 1 what was screened initially (titles, abstracts)? Why 292 papers out of 398 were excluded? What were the reasons to exclude additional 29 papers? Authors have decided to “exclude them with reasons”,  but these reasons are not clarified.   
  2. Table 1 presents all 77 papers extracted for analysis organized in the alphabetical order. I wonder why the Authors have not considered dividing this table into smaller parts, e.g – studies with health professionals (a very special group when it comes to risk of contracting Covid-19) vs. general population or healthy persons vs. those with chronic condition or studies conducted in only one country vs. studies in several countries. It might make the table/tables more readable and informative.
  3. The review might benefit form additional data in Table 1– such as information on how the risk was assessed in a particular study – i.e. specific tools, questions in the survey etc. Without such information it is difficult to understand further presentation of findings, particularly the findings in part 3.4. It is not clear how results from different countries might be compared without previous definition of low/high/ moderate risk and criteria of those. In fact the Authors consider this issue when they state on p. 12: “Given that the studies carried out in Canada, Cyprus, France, Greece, Jordan, Poland, Portugal and Taiwan did not represent the general population or did not mention COVID-19 risk perception scores, they were not discussed within this theme.” It is worth to consider the possible biases of studies and provide a quality score for all studies.  
  4. Page 3 , end of part 2.3. states: “The preliminary quantitative synthesis findings were converted into qualitative themes, which were then pooled together with the results of the initial qualitative synthesis”. It is not clear and more explanation is required.  
  5. It is not clear what was the purpose of the analysis – to find the correlates of higher risk perception or to find the effect of risk perception on individual compliance with pandemic restriction? The statement “These themes were analyzed in relation to four main kinds of COVID-19 risk perceptions: infection, fatality, affective and financial.” (page 26) seems to indicate the purpose but the text does not show that this aim was fulfilled.
  6. Maybe it is worth to look at the theoretical background of analyzed studies? In the Discussion section there are comments related implicitly to Ajzen’s theory or health beliefs model (p. 26). Where these theories applied in any of the studies included in the review?
  7. It seems that the Authors mixed risk of contracting infection and risk of negative (different) consequences of Covid-19 (p. 27). If the studies in the review refer to both that should be clearly stated.
  8. I also found the following sentences in the Discussion section quite strange: “One possible explanation for the heightened threat perceptions among healthcare workers is that when our values are met, both in terms of the importance assigned to them and the perception of living in accordance with them, people are more willing to follow anti-contagion measures during epidemics. In other words, healthcare professionals might present a higher level of COVID-19 risk perception because it is linked to their value of supporting public health, and as a result, they are more likely to engage in protective measures”  (p. 28). Aren’t more protective measures related to higher real risk of contracting COVID-19 in this professional group?

Reviewer 2 Report

Please see the attached PDF file for my comments to the authors. 

Round 2

Reviewer 1 Report

The Authors have included important changes in their manuscript. The amendements have made the text much more clear. I am pleased to notice that most of my comments were considered and addressed properly in the text.